# The Malnutrition Universal Screening Tool (MUST) Predicts Postoperative Declines in Activities of Daily Living (ADL) in Patients Undergoing Cardiovascular Open-Heart Surgery

**DOI:** 10.3390/nu17071120

**Published:** 2025-03-24

**Authors:** Tomomi Oshima, Rie Tsutsumi

**Affiliations:** 1Department of Nutrition, Kikuna Memorial Hospital, Yokohama 222-0011, Japan; 2Department of Nutrition, Dohtai Clinic Kajiwara, Kamakura 247-0063, Japan; 3Department of Anesthesiology and Critical Care, Hiroshima University, Hiroshima 734-8551, Japan

**Keywords:** cardiovascular surgery, malnutrition risk, activities of daily living (ADL), malnutrition universal screening tool (MUST)

## Abstract

***Background*:** Patients undergoing cardiovascular surgery often experience postoperative complications and Activities of Daily Living (ADL) decline, even in the absence of preoperative malnutrition. Since postoperative ADL decline is particularly serious in elderly patients, this study aimed to identify a nutritional assessment tool that is effective in predicting postoperative ADL decline. ***Methods*:** Patients undergoing open-heart surgery with cardiopulmonary bypass were assessed using eight nutritional assessment tools, including the Malnutrition Universal Screening Tool (MUST), the Global Leadership Initiative on Malnutrition (GLIM) criteria, the Nutritional Risk Screening 2002 (NRS-2002), the Subjective Global Assessment (SGA), the Controlling Nutritional Status (CONUT) score, the Prognostic Nutritional Index (PNI), the Geriatric Nutritional Risk Index (GNRI), and the Mini Nutritional Assessment-Short Form (MNA-SF). ***Results*:** A total of 197 patients were enrolled in this study, with a mean age of 70.4 ± 11.6 years old; 31.0% were female. Depending on the nutritional assessment tool, 17.8% to 78.2% of patients were identified as at risk of malnutrition. Among the various assessment tools, the MUST was the most effective in predicting postoperative ADL decline (OR 4.75, 95% CI 1.37–16.5, *p* = 0.014) and was also associated with severe complications and length of hospital stay (OR 3.08, 95% CI 0.20–0.76, *p* = 0.009). On the other hand, nutritional risk detected by assessment tools other than MUST, including MNA-SF and GLIM, could predict postoperative complications but showed no relationship to ADL decline. ***Conclusions*:** We concluded that MUST was the most useful preoperative nutritional assessment tool for predicting outcomes, particularly for assessing the risk of postoperative ADL decline in patients undergoing cardiovascular surgery.

## 1. Introduction

Cardiovascular disease (CVD) is highly prevalent and the most common cause of death worldwide. In particular, the number of patients with aortic and mitral valve diseases has been rapidly increasing due to the aging population in developed countries [1,2]. As a result, the number of cardiovascular surgeries has doubled over the past 30 years in developed countries [3]. In Japan, the number of surgeries has increased approximately fourfold over the past 20 years [4], with about 73,000 cardiovascular surgeries performed annually in 2022 [5]. Malnutrition is widespread in patients with cardiovascular diseases and is associated with adverse postoperative outcomes. Approximately 10–25% of patients undergoing cardiovascular surgery are at risk of nutrition-related issues [6]. Open-heart surgery is highly invasive and has a high incidence of complications, such as postoperative infections and acute renal failure [7,8]. Increased postoperative complications lead to prolonged hospital stays, ADL decline, and increased mortality rates [9]. To prevent these postoperative complications, it is essential to accurately assess nutritional risk and maintain proper nutritional status before surgery. As indicated in the ESPEN guidelines, nutritional support is effective for patients at high nutritional risk [10,11].

The first step in detecting nutritional risk early and providing appropriate nutritional support is nutrition screening and assessment. However, an appropriate nutritional assessment method for patients undergoing cardiovascular surgery has not been established. Several studies have reported that detecting malnutrition through nutritional assessment can predict the incidence of postoperative complications, prolonged hospital stays, and mortality. Nevertheless, the optimal nutritional assessment method remains unclear because the detection rates vary depending on the assessment tool used [12].

Furthermore, factors contributing to preoperative malnutrition include chronic starvation, coexisting inflammatory diseases, age, weight loss over the past six months, low BMI, organ failure, frailty, medications, and more. These diverse factors make it challenging to accurately assess the complex profiles of patients [13]. In addition, there have been no reports to date predicting ADL decline. Therefore, this study aimed to identify the most optimal nutritional assessment tool for patients undergoing cardiovascular surgery, who are prone to postoperative ADL decline. We evaluated the nutritional status of patients undergoing cardiovascular surgery using a variety of existing nutritional assessment tools and examined their relationships with indicators such as postoperative ADL decline and complications.

## 2. Subjects and Methods

### 2.1. Clinical Study Design

This study is a retrospective observation study. This study was conducted in accordance with the Declaration of Helsinki and approved by the Medical Ethics Committee of Kikuna Memorial Hospital (Approval Number: R6-01). The subjects included patients over 20 years old who underwent open-heart surgery with cardiopulmonary bypass in the Department of Cardiovascular Surgery at our hospital between June 2017 and May 2023. The main surgical procedures were valve replacement and coronary artery bypass grafting. Patients who underwent reoperation during the same period were excluded. Preoperative and postoperative baseline assessment items included comorbidities (diabetes, hypertension, chronic kidney disease, dialysis, anemia, heart failure), presence of sarcopenia, and nutritional risk. Nutritional risk was classified into two groups: those with and those without nutritional risk, based on nutritional assessment methods. Postoperative complications were defined as those occurring within 30 days of surgery and classified as Clavien-Dindo Grade III or higher. Changes in ADL were assessed using the Barthel Index, with a postoperative decline compared to preoperative levels defined as a decrease in ADL. The sample size was not calculated due to the retrospective observation study in all patients who met the eligibility criteria.

### 2.2. Nutritional Assessment

Nutritional risk was assessed using the following nutrition assessment tools: MUST (Malnutrition Universal Screening Tool), GLIM (Global Leadership Initiative on Malnutrition), MNA-SF (Mini-Nutritional Assessment-Short Form), NRS-2002 (Nutritional Risk Screening 2002), SGA (Subjective Global Assessment), CONUT (Controlling Nutritional Status), PNI (Prognostic Nutritional Index), and GNRI (Geriatric Nutritional Risk Index). Each assessment was conducted by trained dietitians, and all nutritional assessment data were collected from medical records. The MUST determines nutritional risk based on BMI, weight loss, and the presence of acute disease with inadequate nutritional intake [14]. In this study, nutritional risk was analyzed in two models: one considering both medium risk and high risk to indicate malnutrition, and the other considering only high risk as an indicator of malnutrition. The GLIM rated the risk on two levels. First, all patients were screened using the MNA-SF, which has been validated as a nutrition screening method to identify elderly patients at risk of malnutrition [15]. Next, only patients identified as at risk were further assessed [16]. The muscle loss was assessed using the Skeletal Muscle Index (SMI). In this study, severe, moderate, and mild malnutrition were classified as malnutrition. The MNA-SF, malnutrition and at risk were classified as malnutrition [15]. The NRS-2002, severe, moderate, and mild scores were classified as malnutrition [11]. The SGA, severe and moderate malnutrition were classified as malnutrition [17]. The CONUT, severe, moderate, and mild malnutrition were classified as malnutrition [18]. The PNI score was obtained from serum albumin and total lymphocyte count, and patients with a PNI score of less than 40 were considered to be malnourished [19]. GNRI, moderate, severe, and mild nutritional risks were classified as malnutrition [20].

### 2.3. Physical Assessment

Data on weight, grip strength, cognitive function, weight loss and decreased food intake, blood tests, and biochemical tests were obtained from the medical records of the subjects. Weight was measured using a hospital scale on the day of admission before surgery. The SMI was calculated using a simple estimation formula based on grip strength [21]. Grip strength was measured during the rehabilitation intervention and was measured using an electronic grip strength tester, Grip D (Takei, Niigata, Japan). Grip strength was measured twice on each side by holding the grip strength meter in an upright posture with the arm naturally lowered; the average value of four measurements was calculated, with 0.1 kg as the smallest unit. Data on weight loss and decreased food intake were obtained through interviews with the patient or their family. Blood and biochemical tests data were collected from medical records based on tests performed on the day of admission, the day of surgery, the day after surgery, and several days before discharge.

### 2.4. Statistical Analysis

All statistical analyses were performed with EZR version 1.68 (Saitama Medical Center, Jichi Medical University, Saitama, Japan), which is a graphical user interface for R version 4.4.3 (The R Foundation for Statistical Computing, Vienna, Austria). More precisely, it is a modified version of R commander designed to add statistical functions frequently used in biostatistics. Nutritional risk was evaluated using the following nutritional assessment tools: MUST, GLIM, MNA-SF, NRS-2002, SGA, CONUT, PNI, and GNRI. The patients were divided into two groups: those with malnutrition risk and those without, and postoperative outcomes were compared between the groups. Multivariate analysis was conducted using logistic regression. A *p*-value of <0.05 was considered statistically significant.

## 3. Results

### 3.1. Characteristics of Patients

The patient characteristics are shown in Table 1. After excluding 2 cases that required reoperation, a total of 197 patients were included in the study. The proportion of male patients (69.0%) was higher than that of female patients (31.0%). The average age was 70.4 ± 11.6 years. The mean length of hospital stay was 21.07 ± 36.1 days, and the average BMI was 22.9 ± 4.1. The prevalence of sarcopenia was 25.0%.

### 3.2. Comparison of Nutritional Assessment Methods

Nutritional assessment has often been reported in relation to major outcomes such as mortality and postoperative infections [6,22]. We evaluated nutritional risk using the following nutritional assessment tools: MUST, GLIM, MNA-SF, NRS-2002, SGA, CONUT, PNI, and GNRI. As shown in Table 2, each assessment detects different rates of malnutrition (at nutrition risk, 3.5–21.8%).

Assessment methods that detected nutritional risk frequently included PNI (49.2%), NRS-2002 (severe 21.8%, moderate 13.2%, mild 1.5%), MNA-SF (malnutrition 15.2%, at risk of malnutrition 63.0%), and GNRI (severe 3.6%, moderate 22.3%, mild 11.2%). These methods classified a relatively large number of patients as part of the nutritional risk group. On the other hand, assessment methods that detected relatively few nutritional risks included GLIM (severe 3.5%, moderate 5.6%, mild11.7%), SGA (severe 10.2%, moderate 19.3%), and MUST (high risk 17.8%, medium 14.7%). Furthermore, MUST analyzed nutritional risk in two models: one that considers both medium and high risk as malnutrition and one that considers only high risk as malnutrition. As a result, the percentage of people diagnosed as undernourished differed significantly for each screening tool.

### 3.3. Comparison of ADL Decline and Occurrence of Severe Complications per Nutritional Assessment of Low Nutrition

As shown in Figure 1, the rate of ADL decline was compared between malnutrition patients and patients with no nutritional problems using various nutritional assessment methods. The percentage of patients diagnosed with malnutrition by each nutritional assessment method was different, especially for GLIM, where 20.8% of patients were diagnosed with malnutrition, while 78.2% of patients were diagnosed with malnutrition by MNA-SF. The risk of ADL decline was higher in patients diagnosed with low nutrition by most nutrition assessment methods, but was reversed in GLIM, which found no association between low nutrition and ADL decline. The highest rate of ADL decline was in the group of low-nutrition patients extracted by MUST (20%).

Furthermore, we compared the incidence rates of postoperative complications between malnourished and normal patients for each nutritional assessment method. As shown in Figure 2, MUST showed 37.1% undernourished patients, whereas in the same patients, 14.6% of patients were undernourished when assessed with GLIM.

### 3.4. Tool for Predicting ADL Decline and the Occurrence of Severe Complications

Next, we compared how nutritional risk related to the decline in ADL and the occurrence of severe postoperative complications. Univariate and multivariate analyses were performed for each nutritional assessment method, taking covariates into account. As shown in Table 3, factors predicting the occurrence of ADL decline in the nutritional risk group for each method were examined. Multivariate analysis suggested that the MUST assessment tool was the most effective method of identifying patients deemed to be at medium and high risk of malnutrition, detecting ADL decline at a 4.75-fold higher rate (OR 4.75, CI 1.37 to 16.5, *p* = 0.014). On the other hand, when only high-risk malnutrition patients were evaluated with MUST, there was no significant association with ADL decline (OR 2.38, CI 0.60–9.48, *p* = 0.221). Among other nutritional assessment methods, the univariate analysis showed a trend toward an increased risk of malnutrition in patients and reduced ADL by CONUT (OR 1.22, CI 0.28–2.56, *p* = 1.000), while the multivariate analysis showed a decreased risk for reduced ADL, possibly due to confounding variables (OR 0.42, CI 0.04–4.74, *p* = 0.484). Furthermore, we also investigated the relationship between the nutritional assessment tool used and postoperative complications. As shown in Table 4, MUST, MNA-SF, NRS-2002, SGA, PNI, and CONUT were significantly associated with postoperative complications in the short-variable analysis, while SGA showed a trend toward a decreased risk of postoperative complications in the multivariable analysis, suggesting a confounding variable.

### 3.5. Comparison of Patients with and Without Malnutrition Based on MUST

Table 5 shows the patient background for groups with and without malnutrition, where medium-risk and high-risk categories in the MUST are defined as the malnutrition group. Patients were divided into two groups based on the median BMI value (22.3 kg/m^2^). Compared to the normal group, the malnutrition group had a longer hospital stay (*p* = 0.018), a higher proportion of patients with a BMI below 22.3 (*p* < 0.001), and a higher incidence of severe complications (*p* = 0.009). On the other hand, patients at high risk in the MUST showed a significant correlation with postoperative malnutrition and a higher incidence of severe complications (*p* < 0.001) but no relation to ADL decline (*p* = 0.249) (Table 6).

### 3.6. Relationship Between MUST Subcomponents and ADL Decline

The subcomponents of the MUST are BMI, weight loss, and acute illness with decreased food intake. Among patients who experienced a decline in ADL, the BMI was the factor most frequently considered in patients classified as malnourished based on the MUST. This was consistent both when medium- and high-risk patients were classified as having malnutrition (OR 6.88, *p* < 0.05) and when only high-risk patients were classified as having malnutrition (OR 1.32, *p* < 0.05), as shown in Table 7 and Table 8. Next, a multivariate analysis was performed to determine how the BMI affects the ADL decline (Table 9). The factors considered included length of hospital stay, age, medical history (sarcopenia, diabetes, hypertension), and the BMI. The BMI was classified in two ways: first, based on the median BMI of 22.3 kg/m^2^, where patients were grouped into those with a BMI ≥ 22.3 kg/m^2^ and those with a BMI < 22.3 kg/m^2^; and second, according to the MUST criteria, where patients were grouped into those with a BMI ≥ 18.5 kg/m^2^ and those with a BMI < 18.5 kg/m^2^. The results showed no significant correlation between the ADL decline and BMI when classified according to the median (OR 0.70, CI 0.18–2.72, *p* = 0.610). On the other hand, when classification was performed according to the MUST criteria, there was a significant correlation with the ADL decline (OR 4.86, CI 1.59–14.8, *p* = 0.005).

## 4. Discussion

In this study, we examined nutrition assessment tools for prognostic utility in cardiovascular surgery and showed that MUST nutrition risk was associated with the ADL decline. We examined the nutritional risk of the MUST as both medium and high risk, and the nutritional risk of the MUST as high-risk only, and found that patients at moderate nutritional risk were also at high risk of ADL decline. Although the effectiveness of various nutrition assessment tools in predicting prognosis has been demonstrated, none have reported postoperative ADL decline as an outcome, which is important observation in actual clinical practice.

The MUST is a nutrition screening method that classifies nutritional risk as high-, medium-, or low-risk based on weight loss, dietary intake, BMI, and the presence of acute illness. In a previous study, nutritional risk detected by the MUST preoperatively in patients undergoing open-heart surgery was associated with the occurrence of postoperative complications, prolonged ICU stay, and prolonged hospitalization, suggesting that MUST is effective in predicting prognosis [6,22]. Cardiovascular surgery patients with preoperative weight loss and low BMI are associated with adverse postoperative outcomes, and preoperative nutritional assessment and nutritional therapy are recommended as countermeasures [23]. The present study also suggests that preoperative weight loss and maintenance of reduced dietary intake and BMI can prevent adverse postoperative outcomes and reduce the ADL.

Specific guidelines for preoperative nutritional support in the perioperative period, particularly in cardiovascular surgery, are lacking. The European Society for Clinical Nutrition and Metabolism (ESPEN) recommends preoperative nutritional support 10 to 14 days prior to surgery, and several previous studies have examined the benefits in the field of cardiovascular surgery [24]. However, preoperative nutritional assessment and intervention have not been a priority, even in older patients, as dietary intake is generally good in patients undergoing cardiovascular surgery [25]. Therefore, globally accepted gold-standard nutritional assessment tools for cardiovascular surgery patients are lacking [22]. However, perioperative metabolic changes in cardiovascular surgical patients are not only induced by tissue injury and extracorporeal circulation itself, but also by systemic inflammatory responses to surgical trauma and extracorporeal circulation, perioperative hypothermia, cardiovascular and neuroendocrine responses, and drugs and blood products used to maintain cardiovascular function and anesthesia. Therefore, owing to the association with increased complications, morbidity, and mortality rates, the nutritional status of patients should be evaluated at admission, leading to early recognition of critical patients and the provision of appropriate nutritional support. Compared with patients at low nutritional risk, those at high nutritional risk may benefit more from treatment options [26,27]. The MUST is suggested to not only provide an accurate preoperative nutritional assessment but also prevent a decrease in postoperative intake [28]. This indicates that early postoperative nutritional intervention may also contribute to the recovery of physical function [29,30,31].

In this study, nutritional assessment methods other than the MUST were also examined. Previous studies have shown that GLIM is a predictor of physical decline and mortality in hospitalized patients with cardiovascular disease [32]. GLIM criteria include two etiologic criteria: decreased food intake or absorption and the presence of an inflammatory burden, which, in the case of patients awaiting surgery, as was the case in this study, is characterized by a mild inflammatory response or the absence of an abnormal inflammatory response. Indeed, prior studies have shown that it should be considered in the nutritional assessment of patients with heart disease [33,34]. Several studies have demonstrated that the MNA-SF can accurately assess the risk of malnutrition, quality of life, and functional impairment in patients with heart failure [6,34]. However, its low specificity suggests that it may not be a useful tool in clinical practice [35]. The SGA has been reported to be inaccurate in reflecting true nutritional status, as its anthropometric measurements are influenced by fluid fluctuations, and serum protein levels are affected by inflammatory responses and increased vascular permeability, leading to extravascular leakage [36]. Additionally, for patients awaiting elective surgery, many of the assessment criteria may not be applicable at the time of hospital admission. The NRS-2002 has been significantly associated with an increased incidence of postoperative complications, prolonged ICU stays, and extended hospitalization in patients undergoing cardiac surgery. Therefore, some reports recommend the MUST for its simplicity [37]. When applying the NRS-2002 to cardiac surgery patients, two key limitations should be considered. First, it requires information on patient behavior leading up to surgery and hospitalization; however, recent oral intake and weight loss are not included as assessment criteria. Second, many weight loss evaluation parameters have been reported to be inaccurate due to their susceptibility to fluctuations caused by diuretic therapy for cardiac disease [38]. Patients with a history of cardiovascular disease have been reported to exhibit lower levels of total serum protein, albumin, total cholesterol, triglycerides, and high-density lipoprotein cholesterol, along with higher total white blood cell counts and serum BNP levels. As a result, the CONUT tends to be elevated, while the PNI and the GNRI tend to be lower, potentially leading to relatively higher hazard ratios for adverse outcomes [39,40]. Indeed, it has recently been recognized that serum albumin and prealbumin, known as visceral proteins, characterize inflammation, rather than being positioned as useful biochemical tests in nutritional assessment as in the past [41]. In our study, these were also evaluated in the context of albumin levels, which were largely influenced by inflammation due to surgical invasion, and it is possible that acute inflammation due to surgery or chronic inflammation due to a history of surgery, etc., rather than the impact of undernutrition on postoperative ADL decline, was the major factor in the evaluation. Therefore, a comprehensive nutritional assessment that integrates multiple evaluation methods is necessary and has been suggested to be an effective approach [42].

In this study, the primary outcome was the ADL: as ADL decline, activity decreases, social participation is limited, and home confinement and physical and mental function may increase, leading to decreased independence, the need for care, and ultimately an increased risk of becoming bedridden [43]. In addition, the ADL declines induced decreased mental vitality and increased risk of depression [44]. It has also been reported that decreased ADL may promote undernutrition and chronic heart failure, which may increase hospital stay and mortality [45]. Decreased ADL are directly related to patient QOL after treatment by cardiovascular surgery. Recently, the association between cardiovascular surgery and subsequent QOL has also been recognized as being of great importance [46]. In fact, nearly 40% of patients who have undergone open-heart surgery are not satisfied with their QOL [47]. One of the most relevant factors is the decline in physical function [48]. It will be important to ensure that patients’ QOL does not decline even after successful completion of cardiovascular surgery.

Finally, this study has several limitations. First, it is an observational study, and potential confounding factors may remain due to inadequate adjustment for potential confounders. Second, this is a retrospective study conducted at a single site in Japan, which may limit the results. Therefore, although only BI was used to assess the ADL, the need for more detailed assessment, such as the FIM, and the impact of each screening on the degree of malnutrition should be further investigated. This suggests that further study of nutritional assessment tools for perioperative patients is warranted. The present study demonstrated that MUST predicts a decrease in postoperative the ADL in patients undergoing open-heart surgery.

## 5. Conclusions

This study demonstrated that MUST could predict postoperative the ADL decline in patients undergoing cardiovascular open-heart surgery.

## Figures and Tables

**Figure 1 nutrients-17-01120-f001:**
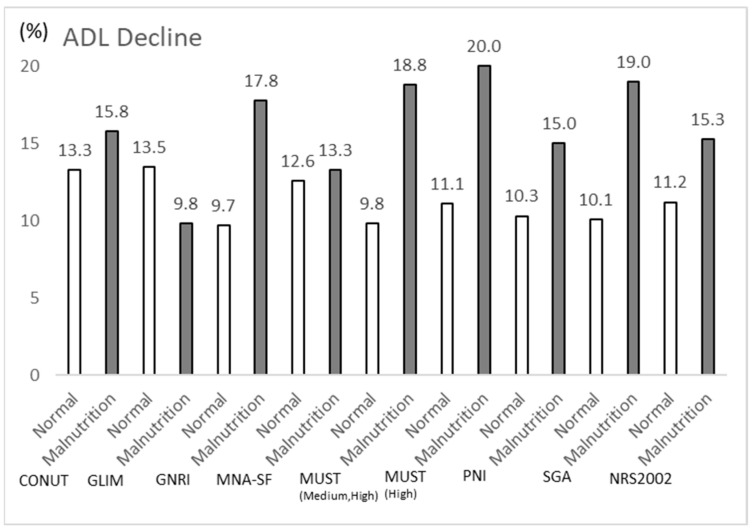
The proportion of malnutrition and ADL decline according to each nutrition screening tool.

**Figure 2 nutrients-17-01120-f002:**
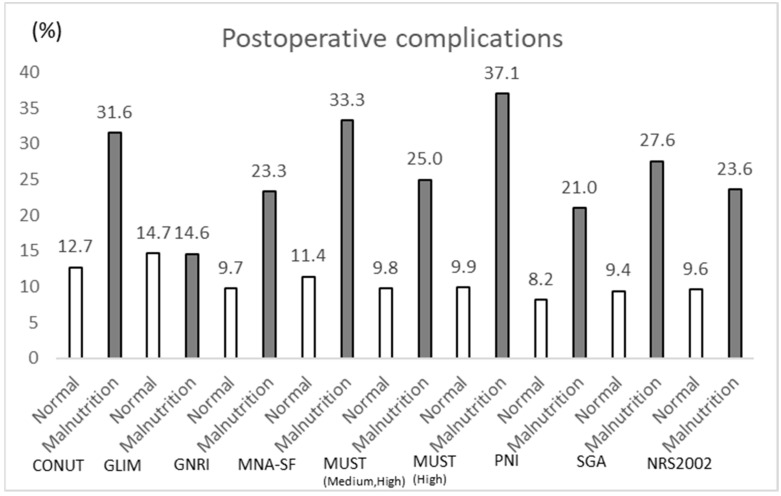
The proportion of malnutrition and the occurrence of severe complications according to each nutrition screening tool.

**Table 1 nutrients-17-01120-t001:** Characteristics of patients.

Factor	N = 197
Age, years (mean ± SD)	70.4 ± 11.6
Female, n (%)	61 (31.0%)
Length of hospital stay (Days) (mean ± SD)	21.07 ± 36.10
Surgical procedure	
Valve Replacement Surgery	141 (71.6%)
Coronary artery bypass grafting	56 (28.4%)
Biochemical parameters	
Albumin, g/dL (mean ± SD)	3.85 ± 0.54
CRP, g/dL (mean ± SD)	1.53 ± 3.38
Anthropometric and body composition characteristics	
BMI, kg/m^2^ (mean ± SD)	22.9 ± 4.1
Barthel Index (mean ± SD)	75.52 ± 31.9
SMI, kg/m^2^, male, n = 92	6.73 (5.27–8.56)
SMI, kg/m^2^, female, n = 47	6.19 (5.16–8.44)
Preoperative handgrip strength below average, n (%)	57 (28.9%)
Preoperative handgrip strength, kg, male, n = 92	29.4 (9.8–53.6)
Preoperative handgrip strength, kg, female, n = 47	18.8 (8.8–37.1)
Sarcopenia, n (%)	89 (45.2%)
Comorbid diseases	
Chronic Kidney Disease, n (%)	20 (10.2%)
Renal function at admission, n (%)	13 (6.6%)
Hemodialysis, n (%)	22 (11.2%)
Diabetes, n (%)	55 (27.9%)
Hypertension, n (%)	106 (53.8%)
Anemia, n (%)	101 (51.2%)

SD: standard deviation; BMI: body mass index; SMI: Skeletal Muscle Index. CRP: C-Reactive Protein.

**Table 2 nutrients-17-01120-t002:** Each nutrition screening/assessment/nutrition evaluation.

Nutritional Assessment	Nutritional Evaluation	N (%)	Nutritional Assessment	Nutritional Evaluation	N (%)
MUST	High risk of malnutrition	35 (17.8%)	SGA	Severely malnourished	20 (10.2%)
	Medium risk of malnutrition	29 (14.7%)		Moderately malnourished or suspected malnutrition	38 (19.3%)
	Low risk of malnutrition	133 (67.5%)		Well-nourished	139 (70.5%)
GLIM Criteria	Severe malnutrition	7 (3.5%)	CONUT	Severe malnutrition	2 (1.1%)
	Moderate malnutrition	11 (5.6%)		Moderate malnutrition	17 (9.2%)
	Mild malnutrition	23 (11.7%)		Mild malnutrition	94 (51.1%)
	Normal Status	156 (79.2%)		Normal nutritional status	71 (38.6%)
MNA-SF	Malnutrition	30 (15.2%)	PNI	PNI ≤ 40	97 (49.2%)
	At risk of malnutrition	124 (63.0%)		PNI > 40	100 (50.8%)
	Malnutrition	43 (21.8%)			
NRS-2002	Severe	43 (21.8%)	GNRI	Severe risk of malnutrition	7 (3.6%)
	Moderate	26 (13.2%)		Moderate risk of malnutrition	44 (22.3%)
	Mild	3 (1.5%)		Mild risk of malnutrition	22 (11.2%)
	Absent	125 (63.5%)		Normal nutritional status	124 (62.9%)

MUST: Malnutrition Universal Screening Tool; GLIM: Global Leadership Initiative on Malnutrition; MNA-SF: Mini Nutritional Assessment-Short Form; NRS-2002: Nutritional Risk Screening 2002; SGA: Subjective Global Assessment; CONUT: Controlling Nutritional Status; PNI: Prognostic Nutritional Index; GNRI: Geriatric Nutritional Risk Index.

**Table 3 nutrients-17-01120-t003:** Nutrition assessment tools associated with ADL decline.

	Univariate Analysis				Multivariate Analysis			
	OR	*p* Value	95% CI		OR	*p* Value	95% CI	
MUST (medium and high risk)	2.13	0.123	0.25	1.08	4.75	0.014	1.37	16.50
MUST (high risk)	2.00	0.249	0.32	1.27	2.38	0.221	0.60	9.48
GLIM	0.69	0.711	0.50	3.80	0.53	0.450	0.10	2.74
MNA-SF	1.07	1.000	0.35	2.55	2.88	0.192	0.59	14.10
NRS2002	1.43	0.545	0.35	1.53	2.08	0.240	0.61	7.05
SGA	2.09	0.140	0.26	1.10	2.03	0.309	0.52	7.94
CONUT	1.22	1.000	0.28	2.56	0.42	0.484	0.04	4.74
PNI	1.54	0.438	0.33	1.46	1.97	0.619	6.30	0.25
GNRI	2.02	0.152	0.50	3.80	1.39	0.419	4.61	0.59

MUST: Malnutrition Universal Screening Tool; GLIM: Global Leadership Initiative on Malnutrition; MNA-SF: Mini Nutritional Assessment-Short Form; NRS-2002: Nutritional Risk Screening 2002; SGA: Subjective Global Assessment; CONUT: Controlling Nutritional Status; PNI: Prognostic Nutritional Index; GNRI: Geriatric Nutritional Risk Index; OR: odds ratio; CI: confidence interval, ADL: Activities of Daily Living.

**Table 4 nutrients-17-01120-t004:** Nutrition assessment tools associated with Postoperative complications.

	Univariate Analysis				Multivariate Analysis			
	OR	*p* Value	95% CI		OR	*p* Value	95% CI	
MUST (medium and high Risk)	3.08	0.009	0.20	0.76	2.85	0.160	0.66	12.30
MUST (high risk)	5.39	<0.001	1.53	4.17	2.29	0.329	0.43	12.10
GLIM	0.99	1.000	0.44	2.31	0.80	0.801	0.14	4.50
MNA-SF	3.89	0.004	0.18	0.66	2.33	0.428	0.29	18.90
NRS2002	2.91	0.014	0.21	0.80	2.20	0.280	0.53	9.23
SGA	3.69	0.002	0.17	0.66	0.59	0.506	0.12	2.82
CONUT	3.16	0.063	0.19	0.87	2.29	0.460	0.25	20.60
PNI	2.96	0.020	0.18	0.84	3.72	0.066	0.92	15.10
GNRI	2.83	0.017	0.21	0.82	1.34	0.677	0.34	5.33

MUST: Malnutrition Universal Screening Tool; GLIM: Global Leadership Initiative on Malnutrition; MNA-SF: Mini Nutritional Assessment-Short Form; NRS-2002: Nutritional Risk Screening 2002; SGA: Subjective Global Assessment; CONUT: Controlling Nutritional Status; PNI: Prognostic Nutritional Index; GNRI: Geriatric Nutritional Risk Index; OR: odds ratio; CI: confidence interval.

**Table 5 nutrients-17-01120-t005:** Factors correlated with MUST (medium and high risk).

MUST (Medium and High Risk)					
Factor	Group	Normal	Malnutrition	*p* Value	SMD
n		133	64		
Barthel Index Decline, n (%)		13 (9.8)	12 (18.8)	0.123	0.259
Postoperative complications, n (%)		13 (9.8)	16 (25.0)	0.009	0.41
Chronic Kidney Disease, n (%)		15 (11.3)	5 (7.8)	0.615	0.118
Hemodialysis, n (%)		15 (11.3)	7 (10.9)	1	0.011
Diabetes, n (%)		43 (32.3)	12 (18.8)	0.069	0.315
Hypertension, n (%)		77 (57.9)	29 (45.3)	0.132	0.254
Sarcopenia, n (%)		60 (59.4)	29 (76.3)	0.098	0.368
Heart failure n (%)		25 (18.8)	16 (25.0)	0.414	0.15
Anemia, n (%)		66 (49.6)	35 (54.7)	0.607	0.101
Body Mass Index, n (%)	22.3 or above	86 (64.7)	13 (20.3)	<0.001	1.004
	below 22.3	47 (35.3)	51 (79.7)		
Surgical procedure, n (%)	Valve Replacement Surgery	94 (70.7)	47 (73.4)	0.815	0.062
	Coronary artery bypass grafting	39 (29.3)	17 (26.6)		
Male, n (%)		96 (72.2)	40 (62.5)	0.226	0.208
Age (mean ± SD)		70.65 (10.45)	69.92 (13.95)	0.684	0.059
Length of hospital stay (mean ± SD)		33.65 (17.78)	41.22 (26.21)	0.018	0.338

SD: standard deviation; SMD: Standardized Mean Difference; MUST: Malnutrition Universal Screening Tool.

**Table 6 nutrients-17-01120-t006:** Factors correlated to MUST (high risk).

MUST (High Risk)					
Factor	Group	Normal	Malnutrition	*p* Value	SMD
n		162	35		
Barthel Index Decline, n (%)		18 (11.1)	7 (20.0)	0.249	0.247
Postoperative complications, n (%)		16 (9.9)	13 (37.1)	<0.001	0.679
Chronic Kidney Disease, n (%)		16 (9.9)	4 (11.4)	1	0.05
Hemodialysis, n (%)		17 (10.5)	5 (14.3)	0.726	0.115
Diabetes, n (%)		46 (28.4)	9 (25.7)	0.91	0.06
Hypertension, n (%)		90 (55.6)	16 (45.7)	0.383	0.198
Sarcopenia, n (%)		75 (62.0)	14 (77.8)	0.299	0.349
Heart failure, n (%)		31 (19.1)	10 (28.6)	0.309	0.223
Anemia, n (%)		80 (49.4)	21 (60.0)	0.341	0.215
Body Mass Index, n (%)	22.3 or above	93 (57.4)	6 (17.1)	<0.001	0.916
	below 22.3	69 (42.6)	29 (82.9)		
Surgical procedure, n (%)	Valve Replacement Surgery	45 (27.8)	11 (31.4)	0.82	0.08
	Coronary artery bypass grafting	117 (72.2)	24 (68.6)		
Male, n (%)		0.72 (0.45)	0.54 (0.51)	0.038	0.375
Age (mean ± SD)		70.29 (11.31)	70.97 (13.38)	0.755	0.055
Length of hospital stay (mean ± SD)		34.12 (17.89)	45.31 (30.83)	0.004	0.444

SD: standard deviation; SMD: Standardized Mean Difference; MUST: Malnutrition Universal Screening Tool.

**Table 7 nutrients-17-01120-t007:** Relationship between MUST (medium- and high-risk) subcomponents and ADL decline.

		OR
ADL decline, n (%)	12 (6.1)	
BMI decreased, n (%)	11 (5.6)	6.88
Weight loss, n (%)	3 (1.5)	0.39
Acute illness and decreased dietary intake, n (%)	1 (0.5)	1.09
BMI and weight loss, n (%)	0 (0)	-
BMI and acute illness and decreased dietary intake, n (%)	0 (0)	-
Weight and acute illness and decreased dietary intake, n (%)	0 (0)	-
All (BMI, weight loss, and acute illness and decreased dietary intake), n (%)	0 (0)	-

BMI: body mass index.

**Table 8 nutrients-17-01120-t008:** Relationship between MUST (high-risk) subcomponents and ADL decline.

		OR
ADL decline, n (%)	7 (3.6)	
BMI decreased, n (%)	7 (3.6)	1.32
Weight loss, n (%)	2 (1.0)	1.09
Acute illness and decreased dietary intake, n (%)	1 (0.5)	1.00
BMI and weight loss, n (%)	1 (0.5)	1.59
BMI and acute illness and decreased dietary intake, n (%)	1 (0.5)	2.38
Weight loss and acute illness and decreased dietary intake, n (%)	0 (0)	0.76
All (BMI, weight loss and acute illness and decreased dietary intake), n (%)	1 (0.5)	2.38

BMI: body mass index.

**Table 9 nutrients-17-01120-t009:** Relationship between BMI and ADL decline.

	Model 1				Model 2			
	OR	*p* Value	95% CI		OR	*p* Value	95% CI	
(Intercept)	0.06	0.095	0.00	1.63	0.04	0.060	0.00	1.15
Length of hospital stay	1.02	0.262	0.99	1.05	1.03	0.072	1.00	1.05
Age	0.99	0.777	0.95	1.04	0.99	0.715	0.95	1.04
BMI	0.70	0.610	0.18	2.72	4.86	0.005	1.59	14.80
Sarcopenia	1.52	0.508	0.44	5.31	1.52	0.509	0.44	5.30
Diabetes	0.94	0.930	0.23	3.91	0.85	0.817	0.21	3.48
Hypertension	1.01	0.989	0.32	3.19	1.13	0.835	0.35	3.62

BMI: body mass index.

## Data Availability

The data presented in this study are available on request from the corresponding author due to ethical restrictions.

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
