# Peer review of "The Malnutrition Universal Screening Tool (MUST) Predicts Postoperative Declines in Activities of Daily Living (ADL) in Patients Undergoing Cardiovascular Open-Heart Surgery"

_nutrients, 2025, doi:10.3390/nu17071120_

Round 1

Reviewer 1 Report

Comments and Suggestions for Authors

COMMENTS FOR AUTHORS

““The Malnutrition Universal Screening Tool (MUST) predicts postoperative declines in activities of daily living (ADL) in patients undergoing cardiovascular open-heart surgery”

by Tomomi Oshima and Rie Tsutsumi

Thank you for submitting your above named manuscript for publication in the Nutrients

SUMMARY

Authors conducted a retrospective study comparing different nutritional assessment tool in order to predict a postoperative decline in activities of daily living in patients undergoing cardiovascular open-heart surgery.

The aim of the paper could be interesting because investigates a real problem with a strong impact on patients’ quality of life and social consequences.

Due to the large number of nutritional tools proposed in literature, I think that the attempt to select the best one for cardiovascular surgery, it could simplify the preoperative assessment, giving some important information on postoperative outcome suggesting some possible action of prehabilitation.

However, the manuscript has been reviewed and is not considered suitable for publication in its present form because includes some mistakes. The reasons for this decision can be found in the  comments reported below.

Table 1.   Sarcopenia  50 pts ( 25%), but  Table 5 reports 89 pats  (60 in Normal group and 29 in Malnutrition one). Table 6 too reports 89 pts.

Table 1 Renal function ad admission 13 pts but the percentage of 66% is wrong.

Author Response

Reviewer #1

We appreciate with your careful peer review and valuable advice. We believe that addressing these comments has improved the quality of the manuscript substantially.

Comment #1;

Table 1.   Sarcopenia  50 pts ( 25%), but  Table 5 reports 89 pats  (60 in Normal group and 29 in Malnutrition one). Table 6 too reports 89 pts.

Response;

We apologize for any confusion caused by our carelessness. The number in Table 1 was a transcription error. The number of 89 sarcopenia patients is correct and Table 1 has been corrected.

Comment #2;

Table 1 Renal function ad admission 13 pts but the percentage of 66% is wrong.

Response;

As with comment 1, this was an inadvertent error on our part. We have corrected it to 13 patients (6.6%).

Reviewer #ï¼’

Comment#1

The methods seem adequate, but some statements need clarification: line 101-102: The PNI, patients were classified based on whether resection or anastomosis was contraindicated. To what do these terms refer? 

The division of patients in two categories is acceptable for the sake of simplicity. However, this causes some loss of information. 

Response#1

We apologize for the confusion caused by our lack of explanation. We have clearly revised the criteria for PNI grouping.

Comment#2

Figure 1: the results of the GLIM score seem at odds with the other score. This should be explained. 

Response#2

Thank you so much for your pointing out. We have added explanation as below and hope that the GLIM results are being adequately discussed and communicated in the discussion.

Line159-167

As shown in Figure 1-A, the rate of ADL decline was compared between malnutirion patients and patients with no nutritional problems using various nutritional assessment methods. The percentage of patients diagnosed as malnutirion by each nutritional assessment method was different, especially GLIM, where 20.8% of patients were malnutirion, while 78.2% of patients were malnutirion by MNA-SF. The risk of ADL decline was higher in patients diagnosed with low nutrition by most nutrition assessment methods, but was reversed in GLIM, which found no association between low nutrition and ADL decline. The highest rate of ADL decline was in the group of low-nutrition patients extracted by MUST (20%).

Comment#3

The data described in line 175 do not correspond well with table 3. On this data hinges the first sentence of the discussion and the conclusion of the article!

Line 177 must be explained: no significant correlation, but with p=0.021. 

Table 3, for CONUT: the odds ration in the univariate analysis is 1.22, in the multivariate analysis, this is 0.42. This also should be explained. 

Response#3

Thank you for your pertinent points, Table3 did not fully explain and caused confusion. We have tried to explain it carefully and have included the following information. We also described about CONUT. Also, the numbers in the table and the text did not match. In fact, when only high risk was considered as low nutritional risk, there was no association with ADL decline. p=0.221 in the table was correct and the numbers in the text were incorrect.

Line 187-197

As shown in Table 3, factors predicting the occurrence of ADL decline in the nutritional risk group for each method were examined. Multivariate analysis suggested that both medium and high risk of malnutrition risk patients by MUST assessment were the most effective assessment tool, detecting ADL decline at a 4.75-fold higher rate (OR 4.75, CI 1.37 to 16.5, P = 0.014). On the other hand, when only high-risk malnutrition patients were evaluated with MUST, there was no significant association with ADL decline (OR 2.38, CI 0.60-9.48, P = 0.221).  Among other nutritional assessment methods, the univariate analysis showed a trend toward an increased risk for malnutrition patients and reduced ADL by CONUT (OR 1.22, CI 0.28-2.56, P = 1.000), while the multivariate analysis showed a decreased risk for reduced ADL, possibly due to confounding variables (OR 0.42, CI 0. 04-4.74, P = 0.484).

Comments #4

The effect of malnutrition (based on MUST) on postoperative complication is an important observation (tables 5 and 6). 

Response #4

We appreciated with your comments. We have described Table5 in more detail as below.

Table 5 showed the patient background for groups with and without malnutrition, where medium risk and high risk categories in the MUST are defined as the malnutrition group. Patients were divided into two groups based on the median value of BMI (22.3kg/m2). Compared to the normal group, the malnutrition group had a longer hospital stay (P = 0.018), a higher proportion of patients with a BMI below 22.3 (P < 0.001), and a higher incidence of severe complications (P = 0.009). On the other hand, patients at high risk in the MUST showed the significant correlation with postoperative malnutrition and a higher incidence of severe complications (P < 0.001), but not related to ADL decline (p=0.249)(Table6).

Comments #4

Table 7 is difficult to read and should be restructured. 

Response #4

Thank you for pointing this out, we have improved Table7 and separated it into Table7 and Table8 for clarity. We have also added Table 9 to explain the relationship between BMI and ADL decline.

Reviewer 2 Report

Comments and Suggestions for Authors

The abstract and the title cover the content very well. The topic is relvant and of interest for the cardiovascular community. The introduction depicts the problem in a relevant way. It also ends with a clear reserach question.

The methods seem adequate, but some statements need clarification: line 101-102: The PNI, patients were classified based on whether resection or anastomosis was contraindicated. To what do these terms refer? 

The division of patients in two categories is acceptable for the sake of simplicity. However, this causes some loss of information. 

Figure 1: the results of the GLIM score seem at odds with the other score. This should be explained. 

The data described in line 175 do not correspond well with table 3. On this data hinges the first sentence of the discussion and the conclusion of the article!

Line 177 must be explained: no significant correlation, but with p=0.021. 

Table 3, for CONUT: the odds ration in the univariate analysis is 1.22, in the multivariate analysis, this is 0.42. This also should be explained. 

The effect of malnutrition (based on MUST) on postoperative complication is an important observation (tables 5 and 6). 

Table 7 is difficult to read and should be restructured. 

Author Response

(The authors gave the same response as above.)

Round 2

Reviewer 2 Report

Comments and Suggestions for Authors

The improvements are significant, but one issue remains to be resolved (table 4):

for SGA, the outcome (OR) for the univariate analysis is difficult to reconciliate with that of the multivariate analysis. 

  • univariate OR=3.69 p=0.002 (0.17-0.66)
  • multivariate: OR=0.59 0.506 (0.12 2.82)

This sfhould be explained 

Author Response

Comment

The improvements are significant, but one issue remains to be resolved (table 4):

for SGA, the outcome (OR) for the univariate analysis is difficult to reconciliate with that of the multivariate analysis.

multivariate: OR=0.59 0.506 (0.12 2.82)

This should be explained

Responseï¼›

Thank you for pointing this out. We have double-checked and believe it may have been caused by confounding or interaction factors rather than a mistake in the statistical analysis. I have added as much explanation for this as possible (Line199-203).

As shown in Table 4, MUST as well as MNA-SF, NRS2002, SGA, PNI, and CONUT were significantly associated with postoperative complications in the short-variable analysis, while SGA showed a trend toward a decreased risk of postoperative complications in the multivariable analysis, suggesting a confounding variable.